# Factors associated with underweight, overweight, and obesity in reproductive age Tanzanian women

Kedir Y. Ahmed[1,2]*, Abdon G. Rwabilimbo[2,3‡], Solomon Abrha[4‡], Andrew Page[2], Amit Arora[2,5,6,7], Fentaw Tadese[8], Tigistu Yemane Beyene[9], Abdulaziz Seiko[10], Abdulhafiz A. Endris[11], Kingsley E. Agho[2], Felix Akpojene Ogbo[2,12], on behalf of the Global Maternal and Child Health Research collaboration (GloMACH)[¶]

1 College of Medicine and Health Sciences, Samara University, Samara, Ethiopia, 2 Translational Health Research Institute, Western Sydney University, Sydney, NSW, Australia, 3 Chato District Council, Geita Region, Northwestern Tanzania, 4 School of Public Health, College of Medicine and Health Science, Wolayta Sodo University, Wolayta Sodo, Ethiopia, 5 School of Health Sciences, Western Sydney University, Sydney, NSW, Australia, 6 Oral Health Services, Sydney Local Health District and Sydney Dental Hospital, NSW Health, Surry Hills, NSW, Australia, 7 Discipline of Child and Adolescent Health, Sydney Medical School, Faculty of Medicine and Health, The University of Sydney, Westmead, NSW, Australia, 8 College of Medicine and Health Sciences, Wollo University, Dessie, Ethiopia, 9 CDT Africa, College of Health Science, Addis Ababa University, Addis Ababa, Ethiopia, 10 CARE Ethiopia, Afar, Ethiopia, 11 Ethiopian Public Health Institute, Addis Ababa, Ethiopia, 12 General Practice Unit, Prescot Specialist Medical Centre Makurdi, Makurdi, Benue State, Nigeria

‡ These authors are joint second authors on this work.
¶ Members of GloMACH are provided in the acknowledgments.
* kedirymam331@gmail.com

## Abstract

### Background

Underweight, overweight, and obesity are major public health challenges among reproductive-age women of lower- and middle-income countries (including Tanzania). In those settings, obesogenic factors (attributes that promote excessive body weight gain) are increasing in the context of an existing high burden of undernutrition. The present study investigated factors associated with underweight, overweight, and obesity among reproductive age women in Tanzania.

### Methods

This study used 2015–16 Tanzania Demographic and Health Survey data (n = 11735). To account for the hierarchical nature of the data (i.e., reproductive age women nested within clusters), multilevel multinomial logistic regression models were used to investigate the association between individual-level (socioeconomic, demographic and behavioural) and community-level factors with underweight, overweight, and obesity.

### Results

Reproductive age women who were informally employed (relative risk ratio [RRR] = 0.79; 95% confidence interval [CI]: 0.64, 0.96), those who were currently married (RRR = 0.59;

**Data Availability Statement:** The analysis was based on the datasets for Tanzania Demographic Health Survey (TDHS). The 2015-16 TDHS datasets can be downloaded [with Measure DHS

approval] from the data page with dataset file name 'TZIR7BDT.ZIP' for STATA https://dhsprogram.com/data/dataset/Tanzania_Standard-DHS_2015.cfm?flag=0. The authors did not have special access privileges.

**Funding:** This study received no grant from any funding agency in public, commercial or not for profit sectors.

**Competing interests:** The authors declare that they have no competing interests.

**Abbreviations:** BMI, Body Mass Index; CI, Confidence Interval; DHS, Demographic and Health Survey; EA, Enumeration Areas; ICF, Inner City Fund; LMICs, Lower and Middle income Countries; MoHCDEC, Ministry of Health Community Development Elderly and children; MoFEA, Ministry of Finance and Economic Affairs; MoCLA, Ministry of Constitution and Legal Affairs; NCD, Non-Communicable Diseases; NIMR, National Institute for Medical Research; OCGS, Chief Government Statistician; RRR, Relative Risk Ratio;SDGs: Sustainable Development Goals; TDHS, Tanzania Demographic and Health Survey; USAID, United State of America Aid; WHO, World Health Organization; MoHCDEC, Ministry of Health Community Development Elderly and children; MoFEA, Ministry of Finance and Economic Affairs; MoCLA, Ministry of Constitution and Legal Affairs.

95% CI: 0.43, 0.82) and those who used contraceptives (RRR = 0.70; 95% CI: 0.54, 0.90) were less likely to be underweight. Reproductive age women who attained secondary or higher education (RRR = 1.48; 95% CI: 1.11, 1.96), those who resided in wealthier households (RRR = 2.31; 95% CI: 1.78, 3.03) and those who watched the television (RRR = 1.26; 95% CI: 1.06, 1.50) were more likely to be overweight. The risk of experiencing obesity was higher among reproductive age women who attained secondary or higher education (RRR = 1.79; 95% CI: 1.23, 2.61), those who were formally employed (RRR = 1.50; 95% CI: 1.14, 1.98), those who resided in wealthier households (RRR = 4.77; 95% CI: 3.03, 7.50), those who used alcohol (RRR = 1.43; 95% CI: 1.12, 1.82) and/or watched the television (RRR = 1.70; 95% CI: 1.35, 2.13).

## Conclusion

Our study suggests that relevant government jurisdictions need to identify, promote, and implement evidence-based interventions that can simultaneously address underweight and overweight/obesity among reproductive age women in Tanzania.

## Introduction

Underweight and overweight (and obesity) have historically been considered as separate public health challenges, but recent global evidence is showing that underweight and overweight/obesity can co-exist within similar communities [1]. This is particularly common in low and middle-income countries (LMICs) where obesogenic factors (attributes that promote excessive body weight gain) are increasing in the context of an existing high burden of undernutrition [2]. The World Health Organization (WHO) describes the coexistence of undernutrition along with overweight, obesity or diet-related non-communicable diseases (NCDs) as the "double burden of malnutrition" (DBM) [3].

In reproductive aged women, the DBM is associated with adverse health and reproductive outcomes. For example, during the early pubertal period, overweight and obesity is associated with psychosocial problems and abnormal uterine bleeding due to irregularity in the menstrual cycle from peripheral conversion of androgens to oestrogen [4–6]. During pregnancy and labour, overweight and obesity are associated with an increased risk of gestational diabetes and pre-eclampsia, haemorrhage, caesarean birthing, and maternal and early neonatal death [7, 8]. In all populations, overweight and obesity are associated with an increased risk of NCDs such as Type 2 diabetes mellitus, cardiovascular and respiratory diseases [9]. In addition, underweight among women is associated with lower economic productivity and can also lead to increased rates of morbidity and mortality [10].

Globally, nearly one-third of the population is affected by at least one form of malnutrition (underweight, overweight, or obesity) [3, 11]. In 2016, more than 600 million adults were underweight, while nearly two billion adults were overweight or obese [12–14]. Evidence has shown that both underweight and overweight are higher among women than men, potentially due to women's reproductive health status, lower social status, poverty, and a lack of education [15–17]. Evidence from LMICs, has suggested that South Asian and sub-Saharan Africa countries (including Tanzania) are facing nutrition and epidemiologic transitions. These mean that there is a shift towards less nutritious and cheaper foods increasing sedentary behaviours, respectively [2, 18–21]. Nutrition and epidemiologic transition has been attributable to changes to available food quantity and quality, rapid economic growth and urbanization [2, 18, 19, 22, 23].

In the past two and half decades in Tanzania, while there has been no reduction in the prevalence of undernutrition, the issues of overweight and obesity are emerging as major public health challenges, particularly in reproductive age women [24]. The Tanzania Demographic and Health Survey (TDHS) report showed that the prevalence of overweight and obesity in reproductive age women increased from 11% in 1991 to 28% in 2016 [24]. However, the prevalence of underweight did not improve over the same period [24]. Previously published studies in Tanzania have shown that overweight or obese reproductive aged women were more likely to be older, educated, married, and resided in wealthier households and urban areas [25, 26]. In contrast, women who were never married, those who resided in poor households, and attained lower education were more likely to be underweight [25, 26]. Although useful, these studies have a number of limitation. First, the studies were based on the older 2010 TDHS data, which may not reflect the current socio-economic, demographic and health situation of the country. Second, the previous studies did not consider the relevant methodological approach (including the hierarchical nature of data) in handling the large dataset. Finally, our study used a statistical modelling technique (multinomial logistic regression and adjusted for relevant confounders), an important statistical approach that was not employed in the past studies conducted in Tanzania.

Understanding of the factors associated with underweight, overweight and obesity among reproductive age Tanzanian women would be helpful for relevant health practitioners to implement evidence based and effective interventions to address both underweight and overweight (and obesity) [27, 28]. This context-specific information is also crucial to national and international stakeholders given the current commitment to achieve Sustainable Development Goal 3 (SDG–3, to end all forms of malnutrition) [29] and the Global Action Plan for the Prevention and Control of NCDs target 9 (to halt the rise in obesity) in Tanzania [30]. Accordingly, the present study investigated factors associated with underweight, overweight, and obesity among reproductive age women in Tanzania.

## Methods

### Data source

The study used the 2015–16 TDHS data (n = 11735). The TDHS was collected by the National Bureau of Statistics, Office of the Chief Government Statistician (OCGS) in Zanzibar and Inner City Fund (ICF) International. The funding for the TDHS was from the Government of Tanzania, Global Affairs Canada and the United States Agency for International Development [24]. The TDHS collected relevant information on maternal and child health indicators, including height and weight measurements for the reproductive age women.

The TDHS used a two-stage stratified cluster sampling technique to select the study participants. In stage one, enumeration areas (EAs) were selected proportional to each geographical zones in Tanzania. The EAs were based on the 2012 Tanzania Population and Housing Censuses [31]. In stage two, a systematic random sampling technique was used to select households after the complete household listing was conducted in each EAs. Out of 13,376 households included in the survey, 13,158 reproductive age women who were permanent residents or who spent the night in the selected households the night before the survey provided weight and height measurements.

For this study, women who were pregnant, or who had given birth in the two months preceding the survey were excluded to minimize measurement bias due to initial weight gain from pregnancy and childbirth, consistent with the TDHS report [24] and previously published studies [18, 25, 26, 32]. A total of sample 11,735 reproductive age women were included in the study. The detailed methodology of the TDHS is provided elsewhere [24].

## Outcome variable

The main outcome variable was the nutritional status of reproductive age women, measured using the WHO adult body mass index (BMI) classification [13], and used by the Tanzanian National Bureau of Statistics and ICF International [24]. BMI was defined as a woman's weight in kilograms divided by the square of her height in meters ($kg/m^2$). The survey used an electronic SECA 874 flat scale designed for mobile use to measure weight and Shorr measuring board to assess height. BMI was classified into four groups, and these subsequently formed the study outcome variables:

- Underweight: BMI < 18.5 $kg/m^2$

- Normal weight: BMI ≥ 18.5 $kg/m^2$ and BMI ≤ 24.9 $kg/m^2$

- Overweight: BMI ≥ 25.0 $kg/m^2$ and BMI ≤ 29.9 kg/m2

- Obesity: BMI ≥ 30.0 $kg/m^2$

## Independent variables

Independent variables were selected based on previous studies from LMICs [18, 19, 25, 26, 32–37] and data availability in the TDHS. These factors were broadly classified as socio-economic, demographic, behavioural and community-level factors.

Socioeconomic factors included women's education (categorised as 'no schooling', 'primary education', or 'secondary or higher education'), women's employment (categorised as 'no employment, 'informal employment or 'formal employment), and household wealth index (categorised as 'poor', 'middle' or 'rich'). Informal employment included agricultural jobs, skilled manual, and unskilled manual works, while formal employment included professional work, technical, management, sales and clerical jobs. Household wealth index was computed by the National Bureau of Statistics and ICF International using principal component analysis (PCA). The PCA considers the ownership of household assets such as toilets, electricity, television, radio, fridge, and bicycle, as well as the availability of a source of drinking water and floor material of the main house [38].

Demographic factors included women's age (categorised as '15–24 years', '25–34 years' or '35 and above years'), women's parity (classified as 'none', '1–4 children' '5 or more children'), and contraceptive use (classified as "Yes" or "No"). Behavioural factors included women's exposure to the media (radio, magazine/newspaper or television), alcohol drinking (classified as "Yes" or "No") and cigarette smoking (classified as "Yes" or "No"). Women who were exposed to the media at least once a week were classified as 'Yes' otherwise were classified as 'No'.

Community-level factors included the place of residence (classified as rural or urban) and region of residence (classified as 'Western Zone', 'Northern Zone', 'Southern Highlands', 'Southern Zone', 'South West Highlands', 'Lake Zone', 'Eastern Zone', 'Central Zone', and 'Zanzibar').

## Statistical analysis

Descriptive analyses involved the calculation of frequencies and percentages for each of the study variables. This was followed by the estimation of the prevalence of underweight, overweight, and obesity by socioeconomic, demographic, behavioural and community-level factors.

Multilevel multinomial logistic regression modelling was used to examine the association between individual-level (socioeconomic, demographic, behavioural factors), and

community-level factors and (i) underweight, (ii) overweight, and (iii) obesity using normal weight groups as a reference category. For this study, using multilevel modelling has the following advantages over the classical single-level logistic regression models. Firstly, multilevel modelling accounts the hierarchical nature of the data (reproductive age women [level I] were conceptualised as being nested within clusters [level II]), consistent with previously published studies [32, 37, 39]. Failing to recognise hierarchical structures underestimates the standard errors of regression coefficients, leading to an overstatement of statistical significance Secondly, multilevel modelling helps to consider the dependence of observations in the same clusters (i.e., reproductive age women in the same cluster tend to be more similar in their nutritional status than in different clusters) [40, 41]. Finally, multilevel modelling allows estimating cluster-level effects (random effects) simultaneously with the measure of associations for community-level predictors (e.g., place of residence).

Regression models were specified in four stages. In stage one, we developed a null unconditional model without any study variable. In stage two, individual-level factors (socio-economic, demographic and behavioural factors) were included in the model. In stage three, community-level factors (place of residence and region of residence) were included without variables from stage two. In the final model, both individual and community-level factors were included in the model. This final model consisting of both the individual and community-level factors model was exported in the results, as it had the smallest deviance and best explains the variation in the outcome variables. S2 Table presented the random variation and model fitness test results of fitted models.

Relative risk ratios with corresponding 95% confidence intervals (CIs) were estimated as the measure of association between the study factors and outcome variables. Multicollinearity was checked using 'vif' command and no significant results were evident. All statistical analyses were conducted using Stata version 14.0 (StataCorp, USA) with 'svy' command to adjust for sampling weights, clustering effects and stratification, and the 'gsem' function was used for multinomial models.

## Ethics approval and consent to participate

The survey was conducted after ethical approval was obtained from the National Institute for Medical Research in Tanzania. During the survey written informed consent were obtained from study participants before the commencement of data collection. Approval to use the data was sought from Measure DHS for this study and permission was granted.

## Results

### Characteristics of the study participants

Among reproductive age women, nearly half (49.6%) were from wealthy households, and 61.6% attained primary education. More than two-fifth (43.7%) of the study participants had no employment, and more than half (53.5%) of women watched the television. Among study participants, 16.3% used alcohol, and 0.4% of them were smokers. Nearly two-third (63.1%) of women resided in rural households (S1 Table).

### Prevalence of underweight, overweight and obesity among reproductive age women in Tanzania

The prevalence of underweight was higher among reproductive age women who had no employment (14.5%) compared to those who were formally employed (5.6%). Women who resided in wealthy households had a higher prevalence of overweight (23.9%) compared to

those who resided in poor-level households (11.5%). The prevalence of obesity was higher among reproductive age women who attained secondary or higher education (13.1%) compared to those who had no schooling (4.9%) [Table 1].

## Factors associated with underweight in Tanzanian women

Reproductive age women who were informally employed were less likely to be underweight compared to those who had no employment (relative risk ratio [RRR] = 0.79; 95% confidence interval [CI]: 0.64, 0.96). Women who were currently married were less likely to be underweight compared to those who were single (RRR = 0.59; 95% CI: 0.43, 0.82). Reproductive age women who used contraceptives were less likely to be underweight compared to their counterparts (RRR = 0.70; 95% CI: 0.54, 0.90). The risk of being underweight was significantly higher among women who smoked cigarettes and those who resided in Southern Tanzania compared to those who did not smoke cigarettes or resided in Western Tanzania, respectively (RRR = 4.31; 95% CI: 1.46, 12.74 for smoking and RRR = 0.33; 95% CI: 0.17, 0.61 for region) [Table 2].

## Factors associated with overweight in Tanzanian women

Reproductive age women who attained secondary or higher education were more likely to be overweight compared to those who had no schooling (RRR = 1.48; 95% CI: 1.11, 1.96), and those who were currently married had a higher risk of being overweight compared to those who were not married (RRR = 1.47; 95% CI: 1.13, 1.91). Women who resided in wealthier households (RRR = 2.31; 95% CI: 1.78, 3.03) and those who reported watching television (RRR = 1.26; 95% CI: 1.06, 1.50) were more likely to be overweight compared to counterparts, respectively. Compared to women ≤ 24 years of age, older women were more likely to be overweight (RRR = 3.31; 95% CI: 2.59, 4.23 for 35–49 years) [Table 2].

## Factors associated with obesity in Tanzanian women

Reproductive age women who attained secondary or higher education had a higher risk of being obese compared to those who had no education (RRR = 1.79; 95% CI: 1.23, 2.61), and formally employed women were more likely to be obese compared to those who had no employment (RRR = 1.50; 95% CI: 1.14, 1.98). Women who were currently married (RRR = 1.78; 95% CI: 1.25, 2.54), and those who resided in wealthier households (RRR = 4.77; 95% CI: 3.03, 7.50) had a higher risk of being obese compared to those who were not married and/or resided in poorer households, respectively. The risk of being obese was higher among older women aged (≥25 years) compared to those who were younger (15–24 years, RRR = 3.92; 95% CI: 2.93, 5.24 for 25–34 years and RRR = 9.94; 95% CI: 7.20, 13.73 for 35–49 years). Women who watched the television were more likely to be obese compared to those who did not watch the television (RRR = 1.70; 95% CI: 1.35, 2.13). The risk of being obese was higher among women who used alcohol compared to those who did not use alcohol (RRR = 1.43; 95% CI: 1.12, 1.82). Women who resided in Northern Zone (RRR = 0.51; 95% CI: 0.34, 0.76) and Lake Zone (RRR = 0.50; 95% CI: 0.33, 0.75) of Tanzania were less likely to be obese compared to those who resided in Western Zone (Table 2).

## Discussion

Our study showed that reproductive age women who were underweight were more likely to be informally employed, currently married, watched the television and resided in the Southern Zone of Tanzania. The risk of being overweight and/or obese was higher among reproductive age women who attained secondary or higher education, resided in wealthy households, were

**Table 1. Prevalence of underweight, overweight and obesity by study factors in reproductive age Tanzanian women, TDHS 2015–16.**

| Variables | Underweight | Normal weight | Overweight | Obesity | **P value |
|---|---|---|---|---|---|
| | *n (%) | n (%) | n (%) | n (%) | |
| **Socioeconomic factors** | | | | | |
| Women's education | | | | | |
| No schooling | 166 (9.9) | 1152 (69.1) | 269 (16.1) | 81 (4.9) | <0.001 |
| Primary school | 663 (9.2) | 4539 (62.8) | 1303 (18.0) | 724 (10.0) | |
| Secondary and above | 277 (9.7) | 1604 (56.4) | 591 (20.8) | 371 (13.1) | |
| Women's employment | | | | | |
| No employment | 394 (14.5) | 1708 (63.0) | 422 (15.6) | 188 (6.9) | <0.001 |
| Formal employment | 52 (5.6) | 416 (44.2) | 269 (28.6) | 203 (21.6) | |
| Informal employment | 171 (6.7) | 1370 (53.6) | 572 (22.4) | 443 (17.3) | |
| Marital status | | | | | |
| Not married | 501 (15.7) | 2118 (66.4) | 437 (13.7) | 133 (4.2) | <0.001 |
| Currently married | 485 (7.0) | 4224 (60.6) | 1409 (20.2) | 848 (12.2) | |
| Formerly married | 121 (7.6) | 953 (60.1) | 317 (20.0) | 195 (12.3) | |
| Household wealth status | | | | | |
| Poor | 494 (12.9) | 2810 (73.2) | 442 (11.5) | 92 (2.4) | <0.001 |
| Middle | 184 (8.8) | 1468 (70.6) | 332 (16.0) | 94 (4.5) | |
| Rich | 428 (7.4) | 3016 (51.8) | 1388 (23.9) | 990 (17.0) | |
| **Demographic factors** | | | | | |
| Women's age | | | | | |
| 15–24 years | 631 (13.4) | 3363 (71.7) | 562 (12.0) | 133 (2.8) | <0.001 |
| 25–34 years | 205 (6.2) | 1987 (60.0) | 724 (21.9) | 395 (11.9) | |
| 35–49 years | 270 (7.2) | 1943 (52.0) | 875 (23.4) | 648 (17.3) | |
| Parity | | | | | |
| None | 491 (16.1) | 2031 (66.4) | 408 (13.4) | 125 (4.1) | <0.001 |
| 1–4 children | 398 (6.8) | 3503 (59.8) | 1196 (20.4) | 759 (13.0) | |
| 5+ children | 217 (7.7) | 1761 (62.3) | 557 (19.7) | 292 (10.3) | |
| **Behavioural factors** | | | | | |
| Listening radio | | | | | |
| No | 309 (12.2) | 1680 (66.1) | 396 (15.6) | 156 (6.1) | <0.001 |
| Yes | 796 (8.7) | 5615 (61.0) | 1767 (19.2) | 1020 (11.1) | |
| Read magazine | | | | | |
| No | 658 (10.1) | 4307 (66.1) | 1079 (16.6) | 473 (7.3) | <0.001 |
| Yes | 447 (8.9) | 2987 (57.2) | 1082 (20.7) | 703 (13.5) | |
| Watch television | | | | | |
| No | 613 (11.3) | 3764 (69.0) | 802 (14.7) | 274 (5.0) | <0.001 |
| Yes | 492 (7.8) | 3530 (56.2) | 1361 (21.7) | 902 (14.4) | |
| Alcohol use | | | | | |
| No | 991 (10.1) | 6181 (62.9) | 1753 (17.8) | 900 (9.2) | <0.001 |
| Yes | 115 (6.0) | 1113 (58.2) | 410 (21.4) | 276 (14.4) | |
| Smoking | | | | | |
| No | 1097 (9.4) | 7264 (62.1) | 2162 (18.5) | 1166 (10.0) | 0.018 |
| Yes | 8 (16.7) | 30 (61.3) | 1 (1.4) | 10 (20.6) | |
| Contraceptive use | | | | | |
| No | 865 (11.6) | 4925 (64.6) | 1199 (16.1) | 581 (7.8) | <0.001 |
| Yes | 241 (5.7) | 2469 (57.80 | 963 (22.6) | 596 (14.0) | |
| **Community-level factors** | | | | | |

*(Continued)*

**Table 1.** (Continued)

| Variables | Underweight | Normal weight | Overweight | Obesity | **P value |
|---|---|---|---|---|---|
| | *n (%) | n (%) | n (%) | n (%) | |
| Place of residence | | | | | |
| Urban | 319 (7.4) | 2213 (52.2) | 1033 (23.9) | 765 (17.7) | <0.001 |
| Rural | 786 (10.6) | 5082 (68.6) | 1129 (15.3) | 411 (5.6) | |
| Region of residence | | | | | |
| Western Zone | 112 (10.4) | 733 (67.7) | 163 (15.1) | 74 (6.8) | <0.001 |
| Northern Zone | 142 (9.9) | 770 (53.8) | 297 (20.8) | 74 (6.8) | |
| Southern Highlands | 41 (7.4) | 500 (68.0) | 128 (17.4) | 223 (15.6) | |
| Southern Zone | 54 (8.4) | 407 (63.2 | 128 (19.9) | 58 (7.9) | |
| Southwest Zone | 43 (4.4) | 712 (65.4) | 214 (19.7) | 111 (10.2) | |
| Lake Zone | 218 (10.1) | 2126 (71.1) | 394 (13.2) | 143 (4.8) | |
| Eastern Zone | 147 (6.6) | 1130 950.5) | 562 (25.1) | 398 (17.8) | |
| Central zone | 122 (14.8) | 738 (63.4) | 198 (17.0) | 53 (4.6) | |
| Zanzibar | 62 (11.9) | 178 (49.2) | 78 (21.7) | 62 (17.2) | |

*n indicates the weighted count

**P value indicates $x^2$ test results in biavariate analysis

currently married and watched television. Women who had formal employment and used alcohol also had higher risk of obesity. Residing in rural households, and Southern Highlands and Lake Zone of Tanzania was associated with lower risk of obesity.

Research indicates that women who are from wealthy households of LMICs have a higher risk of being overweight or obese, while women from wealthy households in higher-income countries have a lower risk of being overweight or obese [42–45]. Our study showed that women who resided in wealthy households were more likely to be overweight or obese compared to those who resided in poor households. The results of this study are consistent with previous studies from South Asian [16, 46] and sub-Saharan Africa countries [18, 34], which showed that women from wealthy households were at a higher risk of overweight or obesity. The likely reason for the relationship between wealthy households, and overweight and obesity could be that women who resided in wealthy households have a reduced level of socio-economical stress and physical activity, and with a less healthy dietary habit (such as poor consumption of fruits and vegetables, and a higher intake of highly caloric foods) compared to those in poor households [47, 48]. The findings of this study suggest that intervention to reduce the burden of overweight or obesity should target on reproductive age women from both middle- and rich-level households.

Global evidence indicates that higher educational attainment is associated with the better health status of the community, due to the improvement in socioeconomic status [49, 50], health literacy and health behaviours [49–51], and self-control and empowerment [50, 51]. This is not always the case in LMIC settings where those with higher education are more likely to be overweight or obese [52, 53]. Consistent with other studies in LMIC settings, we found that reproductive-age women who attained secondary or higher education were more likely to be overweight or obese, similar to studies from Ghana [34], Bangladesh [17], and Ethiopia [18, 19]. Women with higher education are more likely to have a higher socioeconomic status and material resources, and have ready access to energy-dense foods (e.g., sugary drinks) and more sedentary employment (for example, office work) [18, 54]. The perception among the sub-Saharan Africa community (including Tanzania) that a round-body frame as a marker of

**Table 2. Factors associated with underweight, overweight and obesity in reproductive age Tanzanian women, TDHS 2015–16.**

| Variables | Underweight | Overweight | Obesity |
|---|---|---|---|
| | RRR (95% CI) | RRR (95% CI) | RRR (95% CI) |
| **Socioeconomic factors** | | | |
| Women's education | | | |
| No schooling | 1.00 | 1.00 | 1.00 |
| Primary school | 1.22 (0.88, 1.69) | 1.26 (0.97, 1.64) | 1.60 (1.12, 2.28) |
| Secondary and above | 1.16 (0.81, 1.67) | 1.48 (1.11, 1.96) | 1.79 (1.23, 2.61) |
| Women's employment | | | |
| No employment | 1.00 | 1.00 | 1.00 |
| Formal employment | 0.77 (0.55, 1.08) | 1.23 (0.98, 1.53) | 1.50 (1.14, 1.98) |
| Informal employment | 0.79 (0.64, 0.96) | 1.00 (0.85, 1.17) | 1.27 (1.03, 1.58) |
| Marital status | | | |
| Not married | 1.00 | 1.00 | 1.00 |
| Currently married | 0.59 (0.43, 0.82) | 1.47 (1.13, 1.91) | 1.78 (1.25, 2.54) |
| Formerly married | 0.64 (0.41, 0.98) | 1.19 (0.87, 1.64) | 1.56 (1.03, 2.34) |
| Household wealth status | | | |
| Poor | 1.00 | 1.00 | 1.00 |
| Middle | 0.75 (0.57, 1.00) | 1.51 (1.13, 2.02) | 1.89 (1.14, 3.17) |
| Rich | 0.77 (0.60, 1.00) | 2.31 (1.78, 3.03) | 4.77 (3.03, 7.50) |
| **Demographic factors** | | | |
| Women's age | | | |
| 15–24 years | 1.00 | 1.00 | 1.00 |
| 25–34 years | 0.93 (0.69, 1.23) | 2.19 (1.79, 2.68) | 3.92 (2.93, 5.24) |
| 35–49 years | 1.36 (0.94, 1.96) | 3.31 (2.59, 4.23) | 9.94 (7.20, 13.73) |
| Parity | | | |
| None | 1.00 | 1.00 | 1.00 |
| 1–4 children | 0.87 (0.63, 1.20) | 1.19 (0.92, 1.55) | 1.23 (0.86, 1.76) |
| 5+ children | 0.71 (0.45, 1.13) | 1.02 (0.73, 1.42) | 0.99 (0.65, 1.52) |
| **Behavioural factors** | | | |
| Listening radio | | | |
| No | 1.00 | 1.00 | 1.00 |
| Yes | 0.87 (0.70, 1.08) | 1.00 (0.82, 1.21) | 1.03 (0.79, 1.35) |
| Read magazine | | | |
| No | 1.00 | 1.00 | 1.00 |
| Yes | 0.95 (0.79, 1.15) | 1.15 (0.99, 1.33) | 1.33 (1.10, 1.60) |
| Watch television | | | |
| No | 1.00 | 1.00 | 1.00 |
| Yes | 0.90 (0.73, 1.10) | 1.26 (1.06, 1.50) | 1.70 (1.35, 2.13) |
| Alcohol use | | | |
| No | 1.00 | 1.00 | 1.00 |
| Yes | 0.93 (0.66, 1.31) | 1.17 (0.96, 1.31) | 1.43 (1.12, 1.82) |
| Smoking | | | |
| No | 1.00 | 1.00 | 1.00 |
| Yes | 4.31 (1.46, 12.74) | 0.43 (0.90, 2.10) | 1.07 (0.29, 3.95) |
| Contraceptive use | | | |
| No | 1.00 | 1.00 | 1.00 |
| Yes | 0.70 (0.54, 0.90) | 1.12 (0.96, 1.31) | 1.19 (0.98, 1.44) |
| **Community-level factors** | | | |

*(Continued)*

**Table 2.** (Continued)

| Variables | Underweight | Overweight | Obesity |
|---|---|---|---|
| | RRR (95% CI) | RRR (95% CI) | RRR (95% CI) |
| Place of residence | | | |
| Urban | 1.00 | 1.00 | 1.00 |
| Rural | 1.22 (0.99, 1.52) | 0.89 (0.76, 1.06) | 0.70 (0.57, 0.86) |
| Region of residence | | | |
| Western Zone | 1.00 | 1.00 | 1.00 |
| Northern Zone | 0.79 (0.47, 1.31) | 1.08 (0.76, 1.54) | 1.21 (0.81, 1.82) |
| Southern Highlands | 0.64 (0.37, 1.11) | 0.82 (0.57, 1.18) | 0.51 (0.34, 0.76) |
| Southern Zone | 1.00 (0.52, 1.94) | 0.95 (0.64, 1.42) | 1.02 (0.57, 1.81) |
| Southwest Zone | 0.33 (0.17, 0.61) | 1.12 (0.77, 1.62) | 0.72 (0.46, 1.12) |
| Lake Zone | 0.84 (0.52, 1.36) | 0.77 (0.55, 1.09) | 0.50 (0.33, 0.75) |
| Eastern Zone | 0.89 (0.53, 1.49) | 1.12 (0.80, 1.56) | 0.91 (0.62, 1.34) |
| Central Zone | 1.06 (0.64, 1.75) | 0.86 (0.59, 1.25) | 0.61 (0.35, 1.05) |
| Zanzibar | 1.03 (0.64, 1.66) | 1.14 (0.82, 1.59) | 1.27 (0.89, 1.82) |

RRR: Relative risk ratio; 95% CI: confidence interval

socioeconomic success is also the likely explanation for the observed relationship between higher education and overweight and obesity [34, 55]. National policy makers should target on measures and policies that improve physical activity and ensure access to healthy foods by all reproductive-age women.

Increases in women's employment, which is an important determinant of the immediate causes of undernutrition (such as feeding practices and ill-health) and more distal causes of undernutrition (such as income, food security, and education), has a great potential to improve the nutritional status of women [56]. Our study showed that women who were informally employed (in agriculture and manual jobs) had a lower risk of underweight compared to those who had no employment. The negative association between informal employment and underweight can be explained via three pathways [56, 57]. Firstly, the effect of women's employment on their empowerment and household decision making. Secondly, women's employment as a source of income for food and non-food expenditures. Finally, women's employment in agriculture is also a source of food and household consumption.

Moreover, our study showed that women who had formal employment were more likely to be obese compared to those who had no employment. This finding was consistent with previously published studies from 38 LMICs that showed formally employed women were more likely to be overweight or obese [58]. The positive association between formal employment and overweight or obesity can be due to the positive energy balance among formally employed women [58–60]. This positive energy balance may emanate from the less physically active nature of formal jobs (e.g., office works), and also increases in consumption of energy-dense foods (such as sugary drinks) as a result of improvement in the food purchasing power of women [58–60].

Consistent with past studies conducted in Ghana [34], Bangladesh [61], and Myanmar [62], women who watched television had a higher relative risk of overweight or obesity compared to those who did not watch television. Previous studies documented that sedentary behaviours (including watching television) and inadequate physical activity as major risk factors for overweight and obesity [61, 63]. The possible reason for the relationship between watching television and overweight and obesity could be that women who watched television had a reduced

level of physical activity as a result of increased sitting time [61, 63]. In LMICs, having a television can also be considered as a proxy indicator for the higher socioeconomic status of women, which in turn increases the risk of exposure to energy-dense and junk foods [47, 48]. Our study indicated that interventions focused on reproductive age Tanzanian women are required to improve their awareness on the impacts of watching television and sedentary life style.

Social characteristics (including marital status) can influence women's body weight by its moderating effect on diet and physical activity [64, 65]. Our study found that women who were currently married were less likely to be underweight compared to never-married women, but more likely to be overweight or obesity. These findings are consistent with previously published studies conducted in Ethiopia [18], Ghana [34] and Bangladesh [66]. The possible explanation for the relationship between marital status and body weight could be seen in two perspectives [67, 68]. The first perspective related to resources that married women anticipated to have a person with who to eat regularly, which increases the chance to gain bodyweight. The second perspective is based on the attractiveness model that currently married women are less concerned with their body weight as compared to never-married women which are always trying to minimize weight gain to attract a partner.

Our study indicated that reproductive age women who resided in rural households were less likely to be obese compared to those who resided in urban households. The relationship between place of residence and obesity is supported by previously published studies from LMICs [36, 69, 70]. Women from rural households may be engaged in occupational physical activities such as agricultural occupations subject them to labour-intensive activities (manual work) which promotes weight loss and less excess weight gain [58]. Additionally, the reduced chances of rural women to consume processed, packed and refrigerated foods could be the possible explanation for the negative relationship between rural residence and obesity.

## Strengths and limitations

This study has various limitations. First, these findings are limited by the use of cross-sectional data which presents difficulties in establishing a temporal association between the independent variables and the outcome measures. Nevertheless, the observed associations are consistent with studies from other LMICs [18, 34, 61, 62]. Second, BMI does not reflect the location or amount of body fat of women which could be seen as a potential limitation of this study. Despite this, studies have shown that BMI is correlated to more direct measures of body fat, such as underwater weighing and dual-energy x-ray absorptiometry [71]. Third, the study was limited by a lack of data on key factors such as length of time in watching TV, physical activity and total energy expenditure of the urban women, as the TDHS did not collect information on these variables. Fourth, the study factors were measured based on self-report questionnaires which would be a source of measurement bias. Fifth, a lack of subnational assessment given differences across regions may also be considered as a potential limitation. Despite these limitations, the national representativeness of the data and using a standardized questionnaire can be considered as the potential strength of the current study.

## Conclusion

Our study showed varied factors that are associated with the nutritional status of reproductive age women in Tanzania. Underweight was less likely to be evident among women who were informally employed, currently married and used contraception. The risk of being overweight or obese was higher in women who were formally employed, attained secondary or higher education, resided in wealthier households, currently married, and watched television. These

results suggest that there is an increasing need Tanzanian stakeholders to identify, promote, and implement policy interventions that simultaneously address underweight, overweight, and obesity in reproductive age women.

Appropriate intervention strategies focused on reproductive age Tanzanian women–especially on women with risk factors–that promote healthy adult lifestyles (such as physical activity, reducing the intake of sugary drinks, eating fruits and vegetables, and avoiding excessive alcohol consumption) could be implemented to reduce the burdens and impacts of overweight and obesity. In addition, given that Tanzanian women perceive weight gain as an economic success, we propose targeted educational programs that can instil a self-consciousness behaviour on women's weight control. Finally, interventional studies that evaluate the current policy initiatives in addressing underweight, overweight and obesity should be key priorities to improve women's health outcomes.

## Supporting information

**S1 Table. Characteristic of the study participants of reproductive age Tanzanian women, 2015–16.**
(DOCX)

**S2 Table. Factors associated underweight, overweight and obesity among reproductive age women in Tanzania, 2015–16.**
(DOCX)

## Acknowledgments

The authors are grateful to Measure DHS, ICF International, Rockville, MD, USA, for providing the data for analysis. KYA, AGR, KEA, and FAO acknowledge the support of Global Maternal and Child Health Research collaboration.

GloMACH members are Kingsley E. Agho, Felix Akpojene Ogbo, Thierno Diallo, Osita E Ezeh, Osuagwu L Uchechukwu, Pramesh R. Ghimire, Blessing Jaka Akombi, Pascal Ogeleka, Tanvir Abir, Abukari I. Issaka, Kedir Yimam Ahmed, Abdon Gregory Rwabilimbo, Daarwin Subramanee, Nilu Nagdev and Mansi Dhami.

Felix Akpojene Ogbo (f.ogbo@westernsydney.edu.au)–the leader of the collaboration
Kingsley E. Agho, Felix Akpojene Ogbo, Kedir Yimam Ahmed, and Mansi Dhami
Translational Health Research Institute, Western Sydney University, NSW, Australia
Thierno Diallo
School of Psychology, Western Sydney University, NSW, Australia
Osita E. Ezeh, Pramesh R. Ghimire, Tanvir Abir, Abukari I. Issaka, Daarwin Subramanee, and Nilu Nagdev
School of Health Science, Western Sydney University, NSW, Australia
Osuagwu L. Uchechukwu
School of Medicine, Diabetes Obesity and Metabolism Translational Research Unit (DOM-TRU), Macarthur Clinical School, NSW, Australia
Blessing Jaka Akombi
School of Public Health and Community Medicine, Faculty of Medicine, University of New South Wales, NSW, Australia
Pascal Ogeleka
Prescot Specialist Medical Centre, Welfare Quarters, Makurdi, Benue State, Nigeria
Abdon Gregory Rwabilimbo
Chato District Council, Geita Region, Northwestern Tanzania

## Author Contributions

**Conceptualization:** Kedir Y. Ahmed, Abdon G. Rwabilimbo, Solomon Abrha, Felix Akpojene Ogbo.

**Data curation:** Kedir Y. Ahmed.

**Formal analysis:** Kedir Y. Ahmed.

**Methodology:** Kedir Y. Ahmed.

**Software:** Kedir Y. Ahmed.

**Writing – original draft:** Kedir Y. Ahmed, Abdon G. Rwabilimbo, Solomon Abrha.

**Writing – review & editing:** Kedir Y. Ahmed, Abdon G. Rwabilimbo, Solomon Abrha, Andrew Page, Amit Arora, Fentaw Tadese, Tigistu Yemane Beyene, Abdulaziz Seiko, Abdulhafiz A. Endris, Kingsley E. Agho, Felix Akpojene Ogbo.

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
