## [Decision Letter · Decision Letter 0]

20 May 2020

PONE-D-20-06054

Determinants of underweight, overweight, and obesity in reproductive age Tanzanian women: evidence from the 2015–16 demographic and health survey

PLOS ONE

Dear Mr. Ahmed,

Thank you for submitting your manuscript to PLOS ONE. After careful consideration, we feel that it has merit but does not fully meet PLOS ONE’s publication criteria as it currently stands. Therefore, we invite you to submit a revised version of the manuscript that addresses the points raised during the review process.

The manuscript needs to be revised thoroughly for methodological precision. Importantly, the interpretation of the estimates must be reviewed, and causal language should be checked very carefully. Overall, language editing is necessary, all reviewers have remarked substantial shortcomings in this regard.

We would appreciate receiving your revised manuscript by Jul 04 2020 11:59PM. To enhance the reproducibility of your results, we recommend that if applicable you deposit your laboratory protocols in protocols.io, where a protocol can be assigned its own identifier (DOI) such that it can be cited independently in the future. For instructions see: http://journals.plos.org/plosone/s/submission-guidelines#loc-laboratory-protocols

We look forward to receiving your revised manuscript.

Kind regards,

Hajo Zeeb

Academic Editor

PLOS ONE

Journal Requirements:

3. One of the noted authors is a group or consortium [Global Maternal and Child Health Research

collaboration (GloMACH)]. In addition to naming the author group, please list the individual authors and affiliations within this group in the acknowledgments section of your manuscript. Please also indicate clearly a lead author for this group along with a contact email address.

Additional Editor Comments (if provided):

Reviewers' comments:

Reviewer's Responses to Questions

**Comments to the Author**

1. Is the manuscript technically sound, and do the data support the conclusions?

Reviewer #1: Partly

Reviewer #2: Yes

Reviewer #3: Partly

2. Has the statistical analysis been performed appropriately and rigorously? 

Reviewer #1: No

Reviewer #2: Yes

Reviewer #3: No

3. Have the authors made all data underlying the findings in their manuscript fully available?

Reviewer #1: No

Reviewer #2: No

Reviewer #3: Yes

4. Is the manuscript presented in an intelligible fashion and written in standard English?

Reviewer #1: No

Reviewer #2: Yes

Reviewer #3: Yes

5. Review Comments to the Author

Reviewer #1: COMMENTS FOR THE AUTHORS

1) This paper investigated factors associated with underweight, overweight, and obesity using multinomial logistic regressions. The paper adds some data to the literature around the determinants of nutritional status of reproductive-aged Tanzanian women. Notwithstanding, there are some issues that the paper needs to address, inter alia, possible confounders, interpretation of results, and grammar.

2) The authors examined only distal determinants of underweight, overweight, and obesity. Factors and confounders such as food habits of women and other behavioural determinants such as smoking and alcohol consumption that have been consistently considered as proximate determinants of nutritional status were not included in the models. Moreover, the authors on page 15 noted that “poor consumption of fruits and vegetables might be a likely reason…” for the poor nutritional status among women. My advice to the authors is to include these risk factors in the models and further discuss them.

3) Because multinomial logistic regression models were used to investigate the associations, one would expect that the results are interpreted as relative risk ratios (RRR), with reference to the base outcome, in this case, “normal weight.” However, the authors interpreted the results and findings as binary logistic regression, leading to incorrect interpretation of results and contradictory statements throughout the manuscript. For instance, “reproductive-age women who were UNDERWEIGHT were more likely to be currently married”…….” “The odds of OBESITY was higher among currently married women” (page 15). The authors should re-write the entire results and interpret the findings as relative risk ratios (RRR), by comparing the results of underweight, overweight, and obesity to NORMAL WEIGHT. Furthermore, the descriptive results of the “characteristics of the study participants” should be expanded to include other socioeconomic variables.

4) The paper should have a deeper discussion on the status of adult’s health and lifestyle in Tanzania. At the moment, there is no in-depth discussion of the public health policies and intervention programmes targeted at improving the lifestyles and reducing malnutrition. Some further reflection is needed on these issues.

Minor comments

5) The authors need the assistance of an English writer. Many of the points made were obscured by incomplete sentences, difficult phrasing and language. See, for example, page 19,”……. currently married, and watched”. It was a bit difficult for me to follow some of the arguments in the manuscript. Please fix them.

Reviewer #2: In the introduction section first paragraph the word "individuals" does not make sense in the sentence. Likewise, the authors defined double burden instead of showing/describing how LMICs are currently facing a double burden.

The authors have stated, "no previous national-level studies have been conducted to assess the determinants of underweight, overweight, and obesity in reproductive-age women" this is not correct there are other studies in the similar subject matter such as underweight using other terms such as undernutrition as well as obesity using national-level data. The only difference is that they used previous TDHS 2010. It could be more clear for authors to show they have covered that during the literature review and clearly state what will be new in their study. Perhaps using a new model (multinomial), study both underweight, overweight and obesity concurrently using more current (2015-16 TDHS) national data etc.

In the methodology section more explanation on how they reached the sample size of 11,735 is required. From how many households. Did the 11,735 samples account for all reproductive women regardless of whether they have pregnant or not? There should be also further explanations on how they have taken into account the correlation between women/subjects from the same households and sampling weight during analysis.

The authors have stated "the main outcome variables were underweight, overweight, and obesity" in fact this are just categories of nutritional status rather than outcome variables! They should revise this appropriately. To be more clear, authors should use words like exposure, independent, explanatory variables etc rather than study factors! Is there any reason why they did not combine ‘informal employment and ‘formal employment together? I don't see any concrete reason why not to do that.

In the case of results section Table 1 does not add anything crucial relating to study subject matter. Could be more meaningful if combined with the outcome variable of interest i.e. nutrition status categories of overweight, underweight, normal and obese. Therefore, they should combine table 1 and table 2 if necessary.

The first sentence of the conclusion is not clear. Authors should consider revising. The authors have stated "Our study showed that reproductive-age women who were underweight were more likely to be informally employed, currently married, and watched" The word watched does not make sense. I guess perhaps they mean watched television!

Reviewer #3: The manuscripts covers an overall important topic. However, I have some major concerns related to the presentation of results and the discussion.

- First of all, the analysis is based on cross-sectional data (DHS survey). Generall, this is fine, but you cannot claim to analyse "determinants" in this case. Please change it to "Factors associated...".

- The authors stated that there is no previous study on this issue. I seriously doubt this statement. For example, you can refer to a study published on 2019, which is also based on Tanzanian DHS data (10.1186/s12937-019-0505-8).

- The authors claim to have conducted a multilevel analysis. In the abstract it is not clear what these levels are. In the main text it even seems as if it is a hierarchical model. I cannot see the multilevel approach, although individual- and community-level factors have been included in the regression model. This needs to be clarified and maybe even modified with a new analysis.

- Introduction: In how far does the obesogenic environment (which impacts on weight gain) relate to underweight?

- Introduction: Poverty and education directly relate to the social status. In the current formulation it seems as if these are distinguished issues.

- Methods: I suggest to call it "Independent variables" instead of "Study factors".

- Methods: Why did you call the descriptive analysis "preliminary analyses"?

- Methods: I am missing information about the prerequisities of the regression analysis. For example, did you check for multicollinearity? And what is about the model fit?

- Results: The description of the study characteristics is very short.

- Results: I am missing p-values in table 2 to show differences between independent variables and the outcome alread at the bivariate level.

- Results/Discussion: TV watching seems to be associated with underweight, overweight, and obesity. What is the implication of this result? This is only one example for further issues which should be thought during data analysis and interpretation.

- Discussion: Why did you recommend to ensure access to healthy foods only to women with higher educational attainment?

6. PLOS authors have the option to publish the peer review history of their article (what does this mean?). If published, this will include your full peer review and any attached files.

Reviewer #1: No

Reviewer #2: Yes: Edwin Paul

Reviewer #3: Yes: Florian Fischer

---

## [Author Response · Author response to Decision Letter 0]

13 Jul 2020

July 13 2020

Professor Hajo Zeeb

Scientific Editor 

PLOS ONE

Dear Professor Zeeb,

RE: Manuscript resubmission – [PONE-D-20-06054R1] Determinants of underweight, overweight, and obesity in reproductive age Tanzanian women: evidence from the 2015–16 demographic and health survey

Thank you for the invitation to revise our subject-titled manuscript and for the very constructive comments from the reviewers and editor. A revised manuscript (clean and version with track changes) reflecting the following point-by-point response to the reviewers’ comments have been submitted for your consideration.

Academic Editor

General comments

1. The manuscript needs to be revised thoroughly for methodological precision. Importantly, the interpretation of the estimates must be reviewed, and causal language should be checked very carefully. Overall, language editing is necessary, all reviewers have remarked substantial shortcomings in this regard.

Response: 

We thank the editor for the comment and note the methods, results interpretations [as well as casual inference language] and English language in the revised manuscript has been extensively edited by senior authors whose native language is English.

http://www.journals.plos.org/plosone/s/file?id=wjVg/PLOSOne_formatting_sample_main_body.pdf

http://www.journals.plos.org/plosone/s/file?id=ba62/PLOSOne_formatting_sample_title_authors_affiliations.pdf

Response: 

The revised manuscript meets PLOS ONE's style requirements

Response: Response: 

Data availability statement has been revised (Page 23), in line with our previously published manuscript with PLoS One that used similar data [Ahmed et al., 2020]. 

The following text has been added in the revised manuscript

“The analysis was based on the datasets for Tanzania Demographic Health Survey (TDHS). The 2015-16 TDHS datasets can be downloaded [with Measure DHS approval] from the data page with dataset file name ‘TZIR7BDT.ZIP’ for STATA https://dhsprogram.com/data/dataset/Tanzania_Standard-DHS_2015.cfm?flag=0. The authors did not have special access privileges.”

Reference:

1. Ahmed, K. Y., Page, A., Arora, A., Ogbo, F. A., Global, M., & Child Health Research, c. (2020). Associations between infant and young child feeding practices and acute respiratory infection and diarrhoea in Ethiopia: A propensity score matching approach. PLoS One, 15(4), e0230978.

Response: 

As noted above, the data availability statement has been revised (Page 23), in line with our previously published manuscript with PLoS One that used similar data [Ahmed et al., 2020]

Reference:

1. Ahmed, K. Y., Page, A., Arora, A., Ogbo, F. A., Global, M., & Child Health Research, c. (2020). Associations between infant and young child feeding practices and acute respiratory infection and diarrhoea in Ethiopia: A propensity score matching approach. PLoS One, 15(4), e0230978.

4. One of the noted authors is a group or consortium [Global Maternal and Child Health Research collaboration (GloMACH)]. In addition to naming the author group, please list the individual authors and affiliations within this group in the acknowledgments section of your manuscript. Please also indicate clearly a lead author for this group along with a contact email address.

Response: 

Revision done (Page 23 and 24)

Reviewer #1

Comments to the Author

This paper investigated factors associated with underweight, overweight, and obesity using multinomial logistic regressions. The paper adds some data to the literature around the determinants of nutritional status of reproductive-aged Tanzanian women. Notwithstanding, there are some issues that the paper needs to address, inter alia, possible confounders, interpretation of results, and grammar.

Response: 

As noted above and in response to the editor comment, the methods, results interpretations [as well as casual inference language] and English language in the revised manuscript has been extensively edited by senior authors whose native language is English.

The authors examined only distal determinants of underweight, overweight, and obesity. Factors and confounders such as food habits of women and other behavioural determinants such as smoking and alcohol consumption that have been consistently considered as proximate determinants of nutritional status were not included in the models. Moreover, the authors on page 15 noted that “poor consumption of fruits and vegetables might be a likely reason…” for the poor nutritional status among women. My advice to the authors is to include these risk factors in the models and further discuss them.

Response: 

We have some reasons to believe that the reviewer has good knowledge of subject matter – thank you for the comment. We have considered additional variables [based on availability in DHS datasets over the study period] in the revised manuscript as suggested by the reviewer. 

Because multinomial logistic regression models were used to investigate the associations, one would expect that the results are interpreted as relative risk ratios (RRR), with reference to the base outcome, in this case, “normal weight.” However, the authors interpreted the results and findings as binary logistic regression, leading to incorrect interpretation of results and contradictory statements throughout the manuscript. For instance, “reproductive-age women who were UNDERWEIGHT were more likely to be currently married” …….” “The odds of OBESITY was higher among currently married women” (page 15). The authors should re-write the entire results and interpret the findings as relative risk ratios (RRR), by comparing the results of underweight, overweight, and obesity to NORMAL WEIGHT. 

Response: 

Thank you for the critical observation. We apologize for the incorrect interpretation of findings and note that the entire results section has been edited and extensively reviewed by senior authors. 

Furthermore, the descriptive results of the “characteristics of the study participants” should be expanded to include other socioeconomic variables.

Response: 

Revision done (Page 12, Paragraph 1)

The paper should have a deeper discussion on the status of adult’s health and lifestyle in Tanzania. At the moment, there is no in-depth discussion of the public health policies and intervention programmes targeted at improving the lifestyles and reducing malnutrition. Some further reflection is needed on these issues.

Response: 

Thanks for the comment. Revision done (Page 22 paragraph 2)

The authors need the assistance of an English writer. Many of the points made were obscured by incomplete sentences, difficult phrasing and language. See, for example, page 19,”……. currently married, and watched”. It was a bit difficult for me to follow some of the arguments in the manuscript. Please fix them.

Response: 

We apologize for the difficulty caused to the reviewer in reading our manuscript and note [as above] that the language in the revised manuscript has been extensively edited by senior authors whose native language is English.

Reviewer #2

In the introduction section first paragraph the word "individuals" does not make sense in the sentence. 

Response: 

Now revised (Page 4 paragraph 1) 

Likewise, the authors defined double burden instead of showing/describing how LMICs are currently facing a double burden

Response: 

Point appreciated and the situation of DBM in low and middle income countries described (Page 5 paragraph 1)

The authors have stated, "no previous national-level studies have been conducted to assess the determinants of underweight, overweight, and obesity in reproductive-age women" this is not correct there are other studies in the similar subject matter such as underweight using other terms such as undernutrition as well as obesity using national-level data. The only difference is that they used previous TDHS 2010. It could be more clear for authors to show they have covered that during the literature review and clearly state what will be new in their study. Perhaps using a new model (multinomial), study both underweight, overweight and obesity concurrently using more current (2015-16 TDHS) national data etc

Response: 

Revision done (Page 5 and 6).

In the methodology section more explanation on how they reached the sample size of 11,735 is required. From how many households. Did the 11,735 samples account for all reproductive women regardless of whether they have pregnant or not? 

Response: 

Point appreciated and revised accordingly (Page 7, Paragraph 2 and 3)

There should be also further explanations on how they have taken into account the correlation between women/subjects from the same households and sampling weight during analysis.

Response:

In the statistical, sampling weight was account for using the ‘svy’ command – this was noted in the original manuscript.

In our preliminary analyses, we tried to account for the household-level variance, but given the low number of reproductive age women in each households, the dataset could not support inclusion of separate household-level covariates at the household level. We also note that past studies have been conducted on same topic (Tareke et al. 2020, Yeshaw et al., 2020 and Ntenda et al., 2018)

References:

1. Tareke, A. A., & Abate, M. G. (2020). Nutritional paradox in Ethiopian women: Multilevel multinomial analysis. Clinical nutrition ESPEN, 36, 60-68.

2. Yeshaw, Y., Kebede, S. A., Liyew, A. M., Tesema, G. A., Agegnehu, C. D., Teshale, A. B., & Alem, A. Z. (2020). Determinants of overweight/obesity among reproductive age group women in Ethiopia: multilevel analysis of Ethiopian demographic and health survey. 10(3), e034963. doi:10.1136/bmjopen-2019-034963 %J BMJ Open

3. Ntenda, P. A. M., & Kazambwe, J. F. (2018). A multilevel analysis of overweight and obesity among non-pregnant women of reproductive age in Malawi: evidence from the 2015–16 Malawi Demographic and Health Survey. International health, 11(6), 496-506.

The authors have stated "the main outcome variables were underweight, overweight, and obesity" in fact this are just categories of nutritional status rather than outcome variables! They should revise this appropriately. 

Response: 

Point appreciated and now revised (Page 8 paragraph 2)

To be more clear, authors should use words like exposure, independent, explanatory variables etc rather than study factors! 

Response: 

Revision done (Page 8 Paragraph 3)

Is there any reason why they did not combine ‘informal employment and ‘formal employment together? I don't see any concrete reason why not to do that.

Response: 

We appreciate the reviewer’s concern. We, however, note that people who are formally employed (e.g. office workers) may be more likely to have sedentary lifestyle (positive energy balance), particularly in a less developed country like Tanzania with extremely limited availability of physical activity options compared to developed countries like Sweden or Norway with varied options for formally employed workers to attend work. In contrast, informally employed women (e.g., agricultural worker and manual labourers) would have negative energy balance because majority of these women walked hundreds of kilometers weekly to their farm or workplace – in the context of a lack of mechanized agriculture or industrial machineries in a LMIC like Tanzania compared to developed countries. We believe that this classification provided critical information for recommending context-specific strategies for at risk population in Tanzania.

In the case of results section Table 1 does not add anything crucial relating to study subject matter. Could be more meaningful if combined with the outcome variable of interest i.e. nutrition status categories of overweight, underweight, normal and obese. Therefore, they should combine table 1 and table 2 if necessary.

Response: 

Point appreciated and now reflected in the revised manuscript; Table 1 now supplementary table (S1_Table).

The first sentence of the conclusion is not clear. Authors should consider revising. The authors have stated "Our study showed that reproductive-age women who were underweight were more likely to be informally employed, currently married, and watched" The word watched does not make sense. I guess perhaps they mean watched television!

Response: 

Now revised (Page 21 and 22) 

Reviewer #3

The manuscripts covers an overall important topic. However, I have some major concerns related to the presentation of results and the discussion.

Thank you for the comment. The reviewer’s specific comments are addressed below in this rebuttal.

First of all, the analysis is based on cross-sectional data (DHS survey). Generall, this is fine, but you cannot claim to analyse "determinants" in this case. Please change it to "Factors associated...".

Response: 

The text has been revised in the entire manuscript, where applicable.

The authors stated that there is no previous study on this issue. I seriously doubt this statement. For example, you can refer to a study published on 2019, which is also based on Tanzanian DHS data (10.1186/s12937-019-0505-8).

Response:

Now revised (page 5 and 6)

The authors claim to have conducted a multilevel analysis. In the abstract it is not clear what these levels are. In the main text it even seems as if it is a hierarchical model. I cannot see the multilevel approach, although individual- and community-level factors have been included in the regression model. This needs to be clarified and maybe even modified with a new analysis.

Response:

We have revised the text in response to the reviewer comment (page 10 paragraph 2). We note that using multilevel modelling has the following advantages over classical single level logistic regression modelling: 

1. Firstly, multilevel modelling was used to accounts for the hierarchical nature of the data (reproductive age women [level I] were conceptualised as being nested within clusters [level II]), consistent with previously published studies (Tareke et al. 2020, Yeshaw et al., 2020 and Ntenda et al., 2018). Failing to recognise hierarchical structures underestimates the standard errors of regression coefficients, leading to an overstatement of statistical significance 

2. Secondly, multilevel modelling helps to consider the dependence of observations in the same clusters (i.e., reproductive age women in the same cluster tend to be more similar in their nutritional status than in different clusters) (Peugh et al., 2009 and Leyland et al., 2003). 

3. Finally, multilevel modelling allows estimating cluster-level effects (random effects) simultaneously with the measure of associations for community-level predictors (e.g., place of residence).

This text is now included in the revised manuscript. The random effects and the model fitness output of all fitted models also included as supplementary information (S2_Table) for further information on the multilevel modelling fitted.

We would also like to clarify for the reviewer that:

Multilevel modelling is designed to explore and analyse data that come from populations which have a hierarchical and non-hierarchical complex data structures. Hierarchical multilevel modelling works when the lower-level unit nests in one and only one higher-level unit (for example, a child may be nested in only one school) [Snijders et al, 1993]. 

The non-hierarchical multilevel modelling helps to deal with all the different types of designs, realities and research questions that meet ross-classified structures and multiple membership structures. For example, atomic units (individuals) can be nested within more than one unit from a higher-level classification.

In the present study, the hierarchical multilevel modelling was employed to consider the hierarchical nested data structure (i.e., reproductive age women were only nested to a single cluster). Similar analytical strategy was also applied in previously published studies based on demographic health survey data (Tareke et al. 2020, Yeshaw et al., 2020 and Ntenda et al., 2018).

References 

1. Tareke, A. A., & Abate, M. G. (2020). Nutritional paradox in Ethiopian women: Multilevel multinomial analysis. Clinical nutrition ESPEN, 36, 60-68.

2. Yeshaw, Y., Kebede, S. A., Liyew, A. M., Tesema, G. A., Agegnehu, C. D., Teshale, A. B., & Alem, A. Z. (2020). Determinants of overweight/obesity among reproductive age group women in Ethiopia: multilevel analysis of Ethiopian demographic and health survey. 10(3), e034963. doi:10.1136/bmjopen-2019-034963 %J BMJ Open

3. Ntenda, P. A. M., & Kazambwe, J. F. (2018). A multilevel analysis of overweight and obesity among non-pregnant women of reproductive age in Malawi: evidence from the 2015–16 Malawi Demographic and Health Survey. International health, 11(6), 496-506.

4. Peugh JL. A practical guide to multilevel modeling. J Sch Psychol. 2010;48(1):85-112. Epub 2009/12/17. 

5. Leyland AH, Groenewegen PP. Multilevel modelling and public health policy. 2003;31(4):267-74.

6. Snijders, T.A.B., and Bosker, R.J. (1993). Standard errors and sample sizes for two-level research. J. Educational Statist., 18, 237-259.

Introduction: In how far does the obesogenic environment (which impacts on weight gain) relate to underweight?

Response: 

The text has been clarified in the revised manuscript (Page 4 Paragraph 1).

Introduction: Poverty and education directly relate to the social status. In the current formulation it seems as if these are distinguished issues.

Response:

We agree with the reviewer that poverty and education directly relate to the social status. In our study, we have used these variable as presented in the TDHS and used in previously published studies [Rwabilimbo et al., 2020]. We have discussed household wealth and education variables at the beginning of discussion in indicating the importance of these variables (as they are modifiable factors) compared to other social characteristics such as marital status. 

Reference

Rwabilimbo, A. G., Ahmed, K. Y., Page, A., & Ogbo, F. A. (2020). Trends and factors associated with the utilisation of antenatal care services during the Millennium Development Goals era in Tanzania. Trop. Med. Health, 48, 38-38.

Methods: I suggest to call it "Independent variables" instead of "Study factors".

Response: 

Revision done suggested by the reviewer (Page 8 Paragraph 3)

Methods: Why did you call the descriptive analysis "preliminary analyses"?

Response: 

Now revised (Page 10 Paragraph 1)

Methods: I am missing information about the prerequisities of the regression analysis. For example, did you check for multicollinearity? And what is about the model fit?

Response:

Revision done. Reviewers concern reflected in the revised manuscript (Page 10 and 11, S2_Table)

Results: The description of the study characteristics is very short.

Response: 

Revision done (Page 12 paragraph 1). 

Results: I am missing p-values in table 2 to show differences between independent variables and the outcome already at the bivariate level.

Response: 

Revision done as suggested by the reviewer (Table 1)

Results/Discussion: TV watching seems to be associated with underweight, overweight, and obesity. What is the implication of this result? This is only one example for further issues which should be thought during data analysis and interpretation.

Response:

Point appreciated and reflected in the revised manuscript (Page 19 Paragraph 3) 

Discussion: Why did you recommend to ensure access to healthy foods only to women with higher educational attainment?

Response:

The text has been clarified in the revised manuscript (Page 18 Paragraph 1)

We thank the reviewers for the valuable comments and time in reading our manuscript.

We look forward to your final discussion in due course. Please contact me should you require any further information.

Sincerely,

Kedir Yimam Ahmed

Corresponding author

---

## [Editor Report · Decision Letter 1]

24 Jul 2020

PONE-D-20-06054R1

Factors associated with underweight, overweight, and obesity in reproductive age Tanzanian women

PLOS ONE

Dear Dr. Ahmed,

Thank you for submitting your manuscript to PLOS ONE. After careful consideration, we feel that it has merit but does not fully meet PLOS ONE’s publication criteria as it currently stands. Therefore, we invite you to submit a revised version of the manuscript that addresses the points raised during the review process.

The authors have extensively revised the manuscript according to comments made. However, a few issues remain: (Line numbers refer to manuscript with highlighted changes)

L 275 63.1% is not one-third

L279-280 - the highlighting of high or low prevalences according to very different grouping characteristics appears awkward (mentioning smoking and Southwest Tanzania in one sentence, while these categories may overlap). Rather make separate sentences and compare prevalences according to joint characteristics, e.g. by economic status.

L314 Simply state that compared to women≤ 24 years of age, older women were more likely to be overweight. Do not repeat all numbers from the table in the text, but perhaps chose one to exemplify.

Overall, my suggestion is to more strongly group the findings regarding overweight and obesity, as the associated factors are very similar, and in the discussion you are doing this already in most sections. The first discussion section, however, should also be more condensed with regard to overweight/obesity findings.

L 316 check the value 2.192

L.355 check the wording (reported does not fit here)

L. 448 an "a" was added - why?

L 474 Sentence starting "In addition" lacks a verb

We look forward to receiving your revised manuscript.

Kind regards,

Hajo Zeeb

Academic Editor

PLOS ONE

---

## [Author Response · Author response to Decision Letter 1]

25 Jul 2020

July 25 2020

Professor Hajo Zeeb

Scientific Editor 

PLOS ONE

Dear Professor Zeeb,

RE: Manuscript resubmission – [PONE-D-20-06054R2] Factors associated with underweight, overweight, and obesity in reproductive age Tanzanian women

Thank you for the invitation to revise our subject-titled manuscript and for the very constructive comments. A revised manuscript (clean and version with track changes) reflecting the following point-by-point response have been submitted for your consideration.

Comment: L 275 63.1% is not one-third

Response: 

Thank you for the observation! Revision done now (line 250).

Comment: L279-280 - the highlighting of high or low prevalences according to very different grouping characteristics appears awkward (mentioning smoking and Southwest Tanzania in one sentence, while these categories may overlap). Rather make separate sentences and compare prevalences according to joint characteristics, e.g. by economic status.

Response: 

Revision done (Line 255-267) 

Comment: L314 Simply state that compared to women≤ 24 years of age, older women were more likely to be overweight. Do not repeat all numbers from the table in the text, but perhaps chose one to exemplify. 

Response: 

Revision done (Line 288-292) 

Comment: Overall, my suggestion is to more strongly group the findings regarding overweight and obesity, as the associated factors are very similar, and in the discussion you are doing this already in most sections. The first discussion section, however, should also be more condensed with regard to overweight/obesity findings.

Response: 

Revision done (Line 320-326)

Comment: L 316 check the value 2.192

Response: 

Revision done (Line 288-292)

Comment: L.355 check the wording (reported does not fit here)

Response: 

Now revised (Line 326-327)

Comment: L. 448 an "a" was added - why? 

Response: 

Now revised (Line 419) 

L 474 Sentence starting "In addition" lacks a verb

Response: 

Revision done (Line 442-446)

We thank the Editor and reviewers for the valuable comments and time in reading our manuscript.

We look forward to your final discussion in due course. Please contact me should you require any further information.

Sincerely,

Kedir Yimam Ahmed, MPH

Corresponding author

---

## [Editor Report · Decision Letter 2]

3 Aug 2020

Factors associated with underweight, overweight, and obesity in reproductive age Tanzanian women

PONE-D-20-06054R2

Dear Dr. Ahmed,

We’re pleased to inform you that your manuscript has been judged scientifically suitable for publication and will be formally accepted for publication once it meets all outstanding technical requirements.

Kind regards,

Hajo Zeeb

Academic Editor

PLOS ONE
---

## [Editor Report · Acceptance letter]

6 Aug 2020

PONE-D-20-06054R2 

Factors associated with underweight, overweight, and obesity in reproductive age Tanzanian women 

Dear Dr. Ahmed:

I'm pleased to inform you that your manuscript has been deemed suitable for publication in PLOS ONE. Congratulations! Your manuscript is now with our production department. 

Kind regards, 

on behalf of

Prof. Hajo Zeeb 

Academic Editor

PLOS ONE